



# No severe ozone depletion in the tropical stratosphere in recent decades

Jayanarayanan Kuttippurath[1,*], Gopalakrishna Pillai Gopikrishnan[1], Rolf Müller[2], Sophie Godin-Beekmann[3], Jerome Brioude[4]

[1] CORAL, Indian Institute of Technology Kharagpur, Kharagpur–721302, India
[2] Institute of Energy and Climate Research (IEK-7), Forschungszentrum Jülich, Jülich Germany
[3] LATMOS, Sorbonne Université, UVSQ, CNRS, Paris, France
[4] Laboratoire de l'Atmosphère et des Cyclones (LACy), UMR 8105, Météo France/CNRS/ Université de La Réunion, St Denis de La Réunion, France

*Correspondence to*: Jayanarayanan Kuttippurath (jayan@coral.iitkgp.ac.in)

**Abstract.**

Stratospheric ozone is an important constituent of the atmosphere. Significant changes in its concentrations have great consequences for the environment in general and for ecosystems, in particular. Here, we analyse ground-based, ozonesonde and satellite ozone measurements, and reanalysis data to examine the ozone depletion in the tropics, and the spatiotemporal trends in ozone during the past five decades (1980–2020). In the tropics, the amount of column ozone is small (250–270 DU) compared to high and mid-latitudes. In addition, the tropical total ozone trend is very small ($\pm 0$–0.2 DU/yr), compared to high ($\pm 1$–1.5 DU/yr) and mid-latitudes ($\pm 0.75$–1 DU/yr) of both hemispheres as estimated for the period 1998–2018. No measurements and no analyses show any signature of severe stratospheric ozone depletion in the tropics in contrast to a recent claim. Finally, it is very unlikely that an ozone hole would occur outside the Antarctic today with respect to the current stratospheric halogen levels.

## 1 Introduction

Ozone is a triatomic molecule, and 90% of its atmospheric abundance is located in the stratosphere, roughly from 10 t o 50 km above the ground (e.g. Cicerone, 1987; Rowland, 2006; Solomon, 1999). Stratospheric ozone is chemically produced in the tropical stratosphere around 25–35 km and transported to the middle and high latitudes. Therefore, stratospheric ozone mixing ratios are highest in the tropics and decrease towards the polar regions (London, 1992). In general, since the production of ozone is effective in the low latitudes, ozone mixing ratios in the middle and high latitude are smaller than those in the tropics. However, the ozone column, which is the integrated concentration of ozone from the surface to the top of the atmosphere (about 100 km), increases with latitude towards the poles, as the ozone column is determined by atmospheric transport (e.g. Staehelin et al., 2001). As ozone absorbs ultraviolet radiation (UV-B radiation, 280–320 nm), a decrease in its atmospheric



concentration would facilitate more UV incidence on the earth's surface. This is a great concern as UV-B radiation is harmful for life on earth (e.g. Solomon, 1990).

Since the late 1970s, ozone in the Antarctic lower stratosphere has shown a dramatic decrease, which is driven by anthropogenic halogens (Farman et al., 1985; Solomon, 1999). The ozone loss theories, model simulations and measurements (e.g. Tuck et al., 1989; Pyle et al., 1994) showed that the decline in ozone was due to the occurrence of polar stratospheric clouds (PSCs) in winter on which the inactive halogens are converted into active forms, which catalytically destroy ozone in the presence of sunlight during spring (e.g. Solomon et al., 1996; Crutzen and Arnold, 1986; Poole and Mccromick, 1988; Solomon, 1986; 1999). The depletion of ozone deepened in the subsequent years and peaked in the 1990s. The ozone loss in the Antarctic is severe because of the unusual meteorology there, in particular winter/spring periods with very low temperatures and the formation of a polar vortex that effectively isolates the mid-latitude air from polar air. For strong polar ozone loss to occur, it is essential that high levels of active chlorine are maintained up to spring (August, September and October in the Antarctic), allowing a prolonged chlorine activation (Müller et al., 2018).

In contrast, the Arctic temperatures are relatively high, and the polar vortex is disturbed frequently by planetary waves driven by sudden stratospheric warmings (SSWs), which restrict chlorine activation and ozone loss there (e.g. Solomon, 1999) and cause strong year to year variability (von der Gathen et al., 2021). The strong ozone loss in Antarctica led to a situation of column ozone below 220 DU, which is the criterion of an ozone hole, such low values have been present there since the early 1980s to date (Newman et al., 2004; Bodeker et al., 2021; de Laat et al., 2017). The ozone loss in some altitudes in the Antarctic stratosphere reaches the situation of complete loss, which is called the saturation of ozone loss (Hofmann et al., 1991; Kuttippurath et al., 2018). However, ozone loss in other latitude regions, including the Arctic, was not up to the levels in the Antarctic spring. The column ozone loss in the coldest Arctic winter in 2020 was very close to the situation in Antarctica for a few days, but an ozone hole according to the 220 DU criteria never formed (Wohltmann et al., 2020; Grooß and Müller, 2021).

The long-term analysis shows that column ozone loss in the Arctic and Antarctic is relatively severe and the springtime ozone loss in recent years (e.g. 1989–2010) was about 50% and about 5–20%, respectively, for both regions (WMO, 2022). The ozone loss in other seasons was smaller than 10% in the 1990s to date in the polar regions. The southern mid-latitudes had an ozone loss of about 5–7% relative to the 1980 levels, but the loss in the northern mid-latitudes is about half this amount. Conversely, total ozone in the tropics (20°–20° N/S) does not show a clear trend (WMO, 2022). The ozone change in the tropical latitudes is mostly governed by the dynamics related to tropical upwelling rather than by the chemistry as compared to the polar regions.

The change in globally averaged annual total column ozone (TCO) in the mid-1990s with respect to pre-ozone hole (pre-1980) levels is about 5%, but about 17% in Antarctica, and the global TCO remains stable since the 2000s (WMO, 2022; Ball et al.,



2019; Weber et al., 2005). The upper stratospheric decrease in ozone (4–8%) was induced by the increase in chlorine loading from 1980 to the late 1990s (e.g. Steinbrecht et al., 2017), but ozone has been steadily increasing thereafter due to the reduction in stratospheric halogens (WMO, 2018; Harris et al., 2015; Steinbrecht et al., 2017; Godin-Beekmann et al., 2022). The

decrease in upper stratospheric temperature caused by the increase in atmospheric $CO_2$ slows down the ozone loss catalytic reactions, which has also helped to increase ozone there. On the other hand, Bognar et al. (2022) shows a 1−3% reduction in the lower stratospheric ozone since 2000 (Godin-Beekmann et al., 2022). There are also studies indicating a significant reduction in ozone loss rates in Antarctica (Solomon et al., 2016; Kuttippurath and Nair, 2017; Pazmino et al., 2018), but statistically significant positive trends are not detected in other regions (WMO, 2018).

In contrast to the mid-latitudes, ozone loss in the tropics is very small, and available trend analyses also show very small or insignificant values (Thompson et al., 2017, 2021; Randel et al., 1999, 2011; Heue et al., 2016; Lelieveld and Dentener, 2000; Staehelin and Poberaj, 2008, Bognar et al., 2022). However, recently Lu (2022) claimed severe ozone depletion in the tropical stratosphere by using TOST (Trajectory mapped Ozonesonde dataset for the Stratosphere and Troposphere) data for the period 1960–2010. The study claimed that there is even an ozone hole, which is seven times larger than the Antarctic ozone hole.

Furthermore, the ozone hole in the tropics according to that study would be currently increasing and would be a great threat to life in the region.

Chipperfield et al. (2022) showed that there is no robust, credible observational evidence for tropical ozone depletion. Also, the satellite and ground-based observations show that there is only 3–5% decrease in the tropical lower stratospheric ozone, which is far lower than that reported by Lu (2022). Chipperfield et al. (2022) further observe that the number of ozonesonde

profiles used by Lu (2022) is very small, which has an impact on the smoothing method used for generating the TOST data. Since the SHADOZ ozonesonde network was established in the 1990s, there have only been continuous ozone measurements since this period in the southern hemisphere, which are inadequate to claim a year-around large ozone hole in the tropics prior to 1990. Although, the reprocessing has greatly enhanced the ozone data, these profiles were not considered in TOST. Furthermore, the cosmic ray driven electron-induced ozone loss in the tropics and poles are ill-constructed, as it requires

stratospheric clouds, which are not present in the tropical lower stratosphere (Lu, 2010). The CFC-12 observations also do not support the lower stratospheric ozone depletion in the tropics, suggesting that the results of Lu (2022) are flawed. Therefore, we present a thorough investigation of tropical stratospheric ozone and its trend based on various ground-based, satellite and reanalysis data for the past five decades.

## 2 Data and Method

### 2.1 GOZCARDS and SWOOSH ozone profile data sets

Global OZone Chemistry and Related trace gas Data records for the Stratosphere (GOZCARDS v2.2) is a bias-corrected merged satellite-based stratospheric ozone data set for the period since 1979. These data are produced by combining





measurements from different satellites, such as Atmospheric Chemistry Experiment Fourier Transform Spectrometer (ACE-FTS) on SCISAT, Stratospheric Aerosol and Gas Experiment (SAGE) I, SAGE II, Halogen Occultation Experiment (HALOE)

and Microwave Limb Sounder (MLS) on Upper Atmosphere Research Satellite (UARS) and Aura, by using SAGE II data as the primary reference. These data contain ozone mixing ratio and its standard error in the altitude range of 215–0.21 hPa on 10° latitude bins. The GOZCARDS data are in good agreement with other satellite and ground-based ozone measurements, and further details can be found in Froidevaux et al. (2015).

Stratospheric Water and OzOne Satellite Homogenised (SWOOSH) version 2 data are made from the measurements of limb-

sounding satellite instruments; SAGE II, SAGE III, HALOE, UARS MLS and Aura MLS. The primary SWOOSH data are the zonally averaged monthly-mean time series of ozone mixing ratios at pressure levels between 316 and 1 hPa. These data are available from 1980 to date on 2.5°, 5°, and 10° zonal mean grids. The measurements are homogenised by applying the corrections calculated from the measurements taken during the overlap period of those instruments. The bias in different satellite data used for SWOOSH is mostly within 0.2 ppmv with respect to ozonesondes (Davis et al., 2016).

**2.2 SBUV and GSG merged TCO data sets**

The Solar Backscatter Ultraviolet (SBUV) Merged Ozone Data Set (MOD v8.6) provides the longest available satellite-based time series of profile and TCO from a single instrument type for the period 1970–2013 (except a 5-year gap in the 1970s). Data from nine independent SBUV-type instruments are included in the record. Although modifications in instrument design were made in the evolution from the Nimbus-4 Backscattered Ultraviolet instrument to the modern SBUV(/2) type, the basic

principle of measurement and retrieval algorithm remain the same; lending consistency to this data record compared to those based on measurements using different instrument types (Frith et al., 2018). The SBUV zonal mean ozone profiles agree within 10%, mostly within 5%, when compared to ground-based and other satellite measurements (Kramarova et al., 2013; DeLand et al., 2012).

The Merged GOME/SCIAMACHY/GOME2 (GSG) TCO dataset is a comprehensive compilation of measurements from three

satellite instruments: Global Ozone Monitoring Experiment (GOME), Scanning Imaging Absorption Spectrometer for Atmospheric Chartography (SCIAMACHY) and GOME-2 (Lerot et al., 2014). By combining data from multiple instruments, GSG offers improved coverage and temporal continuity from 1995 to date (Weber et al., 2018). This merged dataset provides a consistent and extended record of global TCO. The ozone retrievals are based on the University of Bremen weighing function DOAS (WFDOAS) v4 algorithm (Coldewey-Egbers et al., 2005). These data are in good agreement with the World Ozone





and UV Data Centre (WOUDC) ozonesonde measurements, with an average bias of 2–3% for the zonal and global averaged
        values (Fioletov et al., 2002).

### 2.3 SHADOZ, WOUDC and TOST ozonesonde measurements

Southern Hemisphere ADditional OZonesondes (SHADOZ) is a project designed to measure the vertical profiles of ozone
from a number of tropical stations using ozonesondes, which started in 1998. These measurements make use of Electrochemical

Concentration Cell (ECC) sondes. The ECC instrument has a gas-sampling pump connected to the ozone sensor to a radiosonde
for data telemetry (Komhyr, 1995). The accuracy of ozonesonde measurements are better than 5%. A detailed description of
these data is given in Thompson et al. (2017). Table 1 lists the location of SHADOZ stations.

We also use the WOUDC ECC ozonesonde data for the period 1980–2022. The ECC ozonesonde is interfaced with a
radiosonde, which transmits the data, including ozone, atmospheric pressure, temperature and relative humidity. The

measurements in WOUDC were performed mainly with VIZ radiosondes during the period 1980–1991, followed by RS-80
radiosonde until 2009 and the iMet radiosondes to date. The VIZ radiosondes use a hypsometer for pressure measurements,
and they have an accuracy of ±0.2 hPa at altitudes above 20 hPa (Conover and Stroud, 1958). The RS-80 radiosondes are
paired with electronic boards, which are capable of transmitting data every 7 seconds. However, the advanced V2 versions
have improved electronic components that transmit data every second. The i-Met radiosondes are equipped with a GPS receiver

that measures the geometric altitude, in addition to atmospheric pressure (Johnson et al., 2018).

Trajectory mapped Ozonesonde dataset for the Stratosphere and Troposphere (TOST) is a global 3-dimensional height-
resolved ozone dataset, derived from WOUDC ozone sounding records across the globe using trajectory mapping. These data
are spatially interpolated using 96-hour forward and backward trajectories calculated using the HYSPLIT v 4.8 model at each
1 km altitude from the surface for a number of locations. The National Centers for Environmental Prediction/National Centre

140    for Atmospheric Research (NCEP/NACR) meteorological data are used to drive the trajectory model. A detailed description
of TOST data (variable used: trop_strat_zbith_mean) is given in Tarasick et al. (2019), Liu et al. (2013) and Chipperfield et
al. (2022).

### 2.4 TROPOMI, OMI, OMPS and TOMS ozone column data sets

Tropospheric Ozone Monitoring Instrument (TROPOMI) utilises a combination of spectral bands in the UV and visible

145    wavelength ranges (270–850 nm), specifically designed to capture the absorption features of ozone in earth's atmosphere. By
measuring the intensity of sunlight reflected or scattered by the atmosphere, TROPOMI can retrieve precise information on
the TCO amount. With its high spatial resolution (7 x 5 km), TROPOMI provides global measurements; enabling the
monitoring of ozone distribution, the detection of ozone depletion events, and assessment of ozone trends (Inness et al., 2019).



In general, the retrieval of TCO from TROPOMI employing GODFIT algorithm has an accuracy of about 1% (Spurr et al, 2021).

Ozone Mapping and Profile Suite (OMPS) is one among the five instruments on-board Suomi National Polar-orbiting Partnership (Suomi NPP), which is designed to measure TCO. The spectrometer uses the backscattered solar radiances in each 0.42 nm between 300 to 380 nm, with 1 nm spectral resolution. The swath of OMPS is approximately $50 \times 2800$ km$^2$, with a field of view (FoV) of 0.27° along track and 110° across track with a negative bias of 2–4% compared to reference products (Flynn et al., 2014).

Ozone Monitoring Instrument (OMI) has been key in providing accurate measurements of TCO from 2004 onwards. By applying the advanced Ultraviolet (UV) and Visible (VIS) spectrometry techniques, OMI captures sunlight scattered by Earth's atmosphere to determine ozone concentrations (Levelt et al., 2006). It operates in two wavelength ranges: 270–370 nm in UV and 350–500 nm in VIS. The spectral resolution is 0.45 nm for UV and 0.63 nm for VIS. Its retrieval algorithm (DOAS) processes the spectral information to derive TCO values. Its high spatial resolution (25 x 25 km) enables detailed mapping of global distribution of TCO with a bias less than 6% in the tropics and mid-latitudes (Huang et al., 20187).

Total Ozone Mapping Spectrometers (TOMS) are a series of instruments designed to measure TCO. Here, we use TCO measurements from TOMS aboard Nimbus-7 (N7) and Earth Probe (EP) covering the period from 1979 to 2004 (McPeters et al., 1998). TOMS employs a single monochromator and a scanning mirror to sample the backscattered solar ultraviolet radiation at 3° intervals along a line perpendicular to the orbital plane. EP-TOMS employs six discrete wavelengths ranging from 309 to 360 nm, using triangular slit functions with a nominal 1 nm bandwidth. The estimated uncertainty of TOMS data is about 3.3%, and there is a bias of 1–2% among the ozone data from different TOMS platforms (Kroon et al., 2008).

### 2.5 MERRA-2 and ERA 5 Reanalyses

MERRA–2 (Modern Era Retrospective analysis for Research and Analysis–2) is a reanalysis data developed at the Global Modelling and Assimilation office (GMAO). These data have 72 sigma pressure vertical layers with a spatial resolution of 0.625° $\times$ 0.5° (Bosilovich et al., 2015) and are available from 1980 onwards. The reanalysis also assimilates data from Infrared Atmospheric Sounding Interferometer (IASI) and Cross Infrared Sunder and Advanced Technology Microwave Sounder on board Suomi NPP. An adaptive bias correction scheme is also applied to the radiance data while assimilating the datasets in MERRA–2. A comprehensive discussion on this assimilation system is given in Gelaro et al. (2016).

We have also used the ECMWF Reanalysis 5[th] Generation (ERA5) data, which has 137 vertical hybrid sigma pressure levels, which extends from the surface to 0.01 hPa. We use the monthly gridded data at a 0.25° $\times$ 0.25° resolution for the period 1984–2018 (Hersbach et al., 2019). Here, the ozone mass mixing ratio (MMR) in kg/kg is converted to parts per million (ppm) by multiplying it with the dividend of molar mass of dry air (28.9644) and molar mass of ozone (47.9982).





We have estimated the long-term trends in ozone by applying the linear method using two sets of measurements. It defines a two-sided alternative hypothesis, for which the slope of regressed line is non-zero. The standard error of the slope is estimated using the assumption of residual normality, and statistical significance of the trend is estimated by finding the $p$-value derived from the Wald-Test with t-distribution. We have considered the slope to be statistically significant if its p-value is $< 0.05$ (95% CI).

## 3. Results and Discussion

### 3.1 Ozone variability and trends across the latitudes

Figure 1 shows the latitudinal distribution of (zonal mean) stratospheric ozone from a merged satellite dataset (GOZCARDS). The data show high ozone mixing ratios (10–11 ppm at 25–35 km) in the tropics (30° N–30° S), which decrease toward the high latitudes (2–5 ppm at 25–35 km). Since the production of ozone is effective in the tropics, ozone mixing ratios are highest
in the tropical middle stratosphere (25–35 km). As the intensity of atmospheric transport is different with seasons, there are also analogous changes in ozone distribution across the latitudes and altitudes. The seasonal variability of ozone is minimum in the tropics and very high in the polar regions with respect to the latitudinal distribution of sunlight and variability of the dynamical processes. Therefore, the seasonal averages show comparatively high ozone in summer and spring, and relatively lower ozone in autumn and winter in the tropics. Since the winter transport is stronger, the ozone values in the northern
hemispheric middle and high latitudes are comparatively higher during this period (e.g. see the 4 ppm contour). Relatively lower ozone values are found in the mid-latitudes (e.g. 6–7 ppm at 10 hPa), but lowest in the polar regions (3–4 ppm at 100 hPa). The smaller wintertime ozone values in the polar lower stratosphere (1–3 ppm) indicate the seasonal ozone loss there (e.g. Randel and Cobb, 1994; Weatherhead and Andersen, 2006; Chipperfield et al., 2015; Muller et al., 2008; von der Gathen et al., 2021).



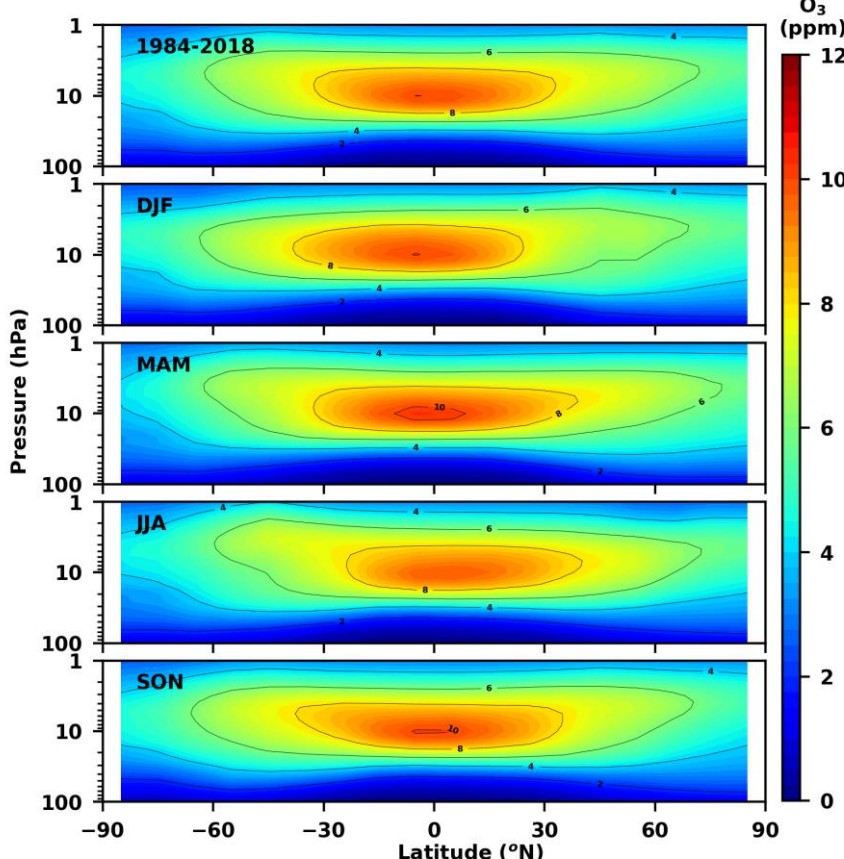


Figure 1: Latitudinal distribution of ozone mixing ratios in ppm averaged from 1984 to 2018 and throughout the seasons, as analysed from the GOZCARDS data. Here, DJF is December, January and February; MAM is March, April and May; JJA is June, July and August; SON is September, October and November.

Figure 2 shows the seasonal distribution of TCO across the latitudes for 2015 measured by OMPS. In contrast to mixing ratios,
the TCO distribution shows high values in the northern high latitudes in winter and spring, and very low values in the southern hemisphere spring and autumn. The Antarctic ozone hole is clearly visible in austral spring, but the analysis for the boreal spring is masked by the data gaps. However, a reduction of around 50–60 DU, which is the average TCO loss expected in a normal cold Arctic winter, in 0°–50° E and 100°–130° E around 70° N, is clearly captured (Goutail et al., 2005). The seasonal variation of ozone in the tropical latitudes is very small, but the southern mid-latitudes show high values in winter, and northern
hemisphere in spring, as the Brewer Dobson Circulation (BDC) is stronger in winter and spring (Lin and Fu, 2013; Flury et al., 2013; Fu et al., 2010). Here, we have used the data from OMPS for the year 2015 to show the changes in TCO, since there was a pronounced Antarctic ozone hole in that year.



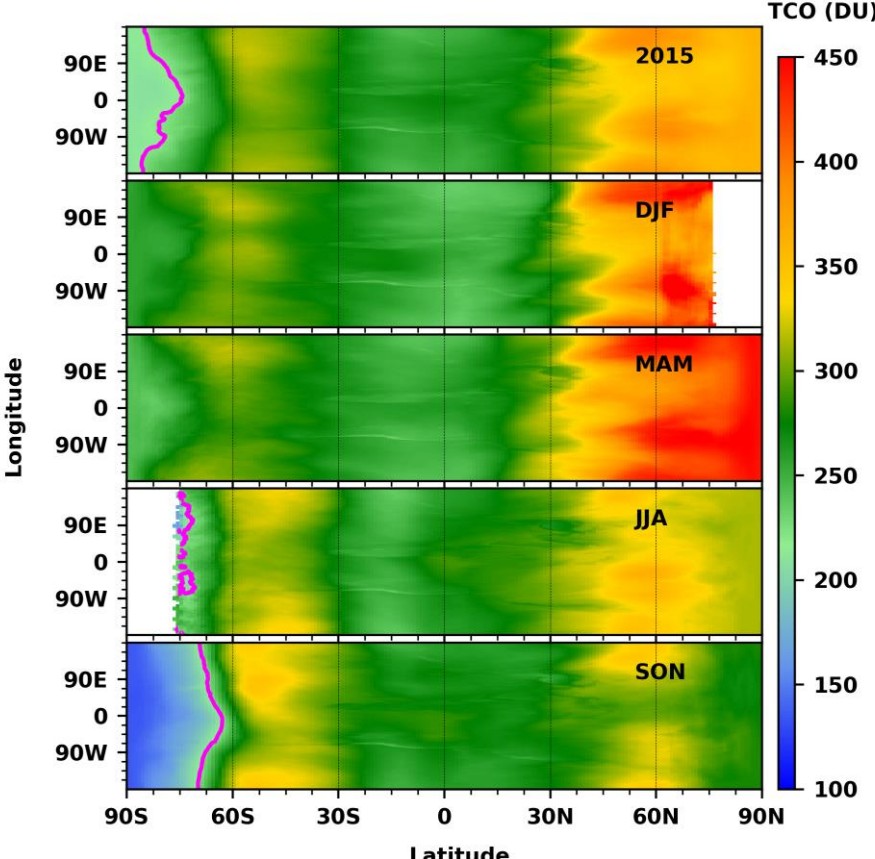

Figure 2: The global seasonal and annual distribution of Total Column Ozone (TCO in DU) for 2015 as measured by Ozone Mapping and
Profiler Suite (OMPS). The magenta lines show the region of ozone hole, i.e., less than 220 DU. The white spaces are data gaps. Here, DJF
is December, January and February; MAM is March, April and May; JJA is June, July and August; SON is September, October and
November.

Figure 3 shows the TCO averaged over the tropics for the period 1978–2022, which is within 250–280 DU over this time
period from all available measurements and reanalysis datasets. We also observe a decrease in peak TCO in the tropics during
the period 1995–1999 (around 255 DU) when compared to the previous and following years (> 255 DU). However, there is
an increase in TCO post–1997 and there is no significant difference (10–15 DU) in TCO among different data sets during the
entire period. Furthermore, the bias in measurements from different instruments is within 5–10 DU, which shows that the data
are robust and there is no substantial loss of ozone in the tropics. The tropical column ozone is never below 220 DU. We have
also computed the trends in TCO using MERRA-2, ERA-5 and satellite data (combined SBUV and OMPS measurements).
The satellite-based estimates show significant negative trends (-0.076±0.028 DU/yr and -0.093±0.059 DU/yr) in the pre- and
post-1997 periods, whereas the reanalysis data show nonsignificant trends in both periods. Conversely, the GSG (GOME–
SCIAMACHY–GOME 2) data yield nonsignificant positive trends during the post-2000 period.





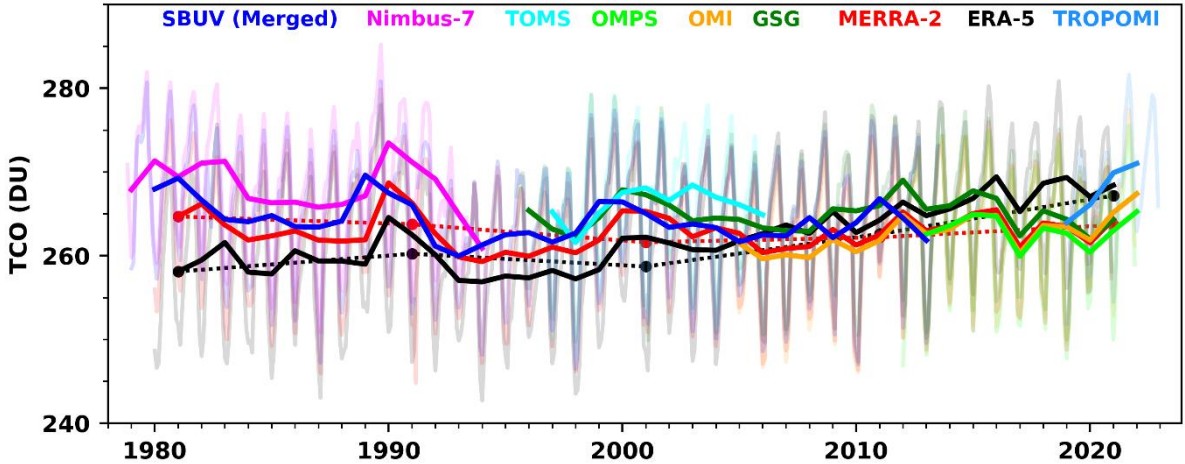

Figure 3: The distribution of Total Column Ozone (TCO in DU) averaged over the tropics (30° S–30° N) from different satellites from 1978
to 2022. The light lines show the monthly distribution, whereas dark lines show the annually averaged value of TCO. The dotted line shows
the decadal distribution of TCO from MERRA–2 and ERA–5. Here, TOMS is Total Ozone Mapping Spectrometer, OMPS is Ozone
Mapping and Profiler Suite, OMI is Ozone Monitoring Instrument, TROPOMI is Tropospheric Ozone Monitoring Instrument and GSG is
Global Ozone Monitoring Experiment (GOME) - SCanning Imaging Absorption spectroMeter for Atmospheric CHartographY
(SCIAMACHY) - GOME-2 satellite data. MERRA-2 is Modern-Era Retrospective analysis for Research and Applications, version 2
reanalysis and ERA5 is the 5[th] Generation European Centre for Medium-Range Weather Forecasts (ECWMF) atmospheric reanalysis data
set. The peak in 1991 may be driven by the Mount Pinatubo volcanic eruption.

We have estimated the trends in ozone in the stratosphere using the SWOOSH (Fig. 4) and GOZCARDS (Fig. S1) data for the
period 1984–2018, and the trends are statistically nonsignificant (at the 95 CI) at most altitudes for both data sets. The
SWOOSH estimates for the period 1984–1997 show high statistically nonsignificant negative trends, about 0.03–0.04 ppm/yr
in the upper stratosphere and 0.01–0.02 ppm/yr in the middle and lower stratosphere. Some regions also show nonsignificant
positive trends (0.03–0.04 ppm/yr) such as the tropical lower stratosphere in all seasons (but DJF and JJA in GOZCARDS and
these are significant). The negative trends indicate the impact of high amounts of stratospheric halogens during the period
1984–1997. In contrast, the estimates for the period 1998–2018 show nonsignificant positive trends (0.01–0.025 ppm/yr)
throughout the stratosphere across the seasons, except in the high-latitude lower stratosphere, where the values are slightly
negative (around 0.01 ± 0.002 ppm/yr) and significant. The positive trends in other latitudes and altitudes are mostly within
0.01–0.02 ppm/yr, and are significant. The highest among these trends (0.025 ± 0.01 ppm/yr) are found in the northern and
southern low-latitude mid stratosphere (above 10 hPa) in MAM.



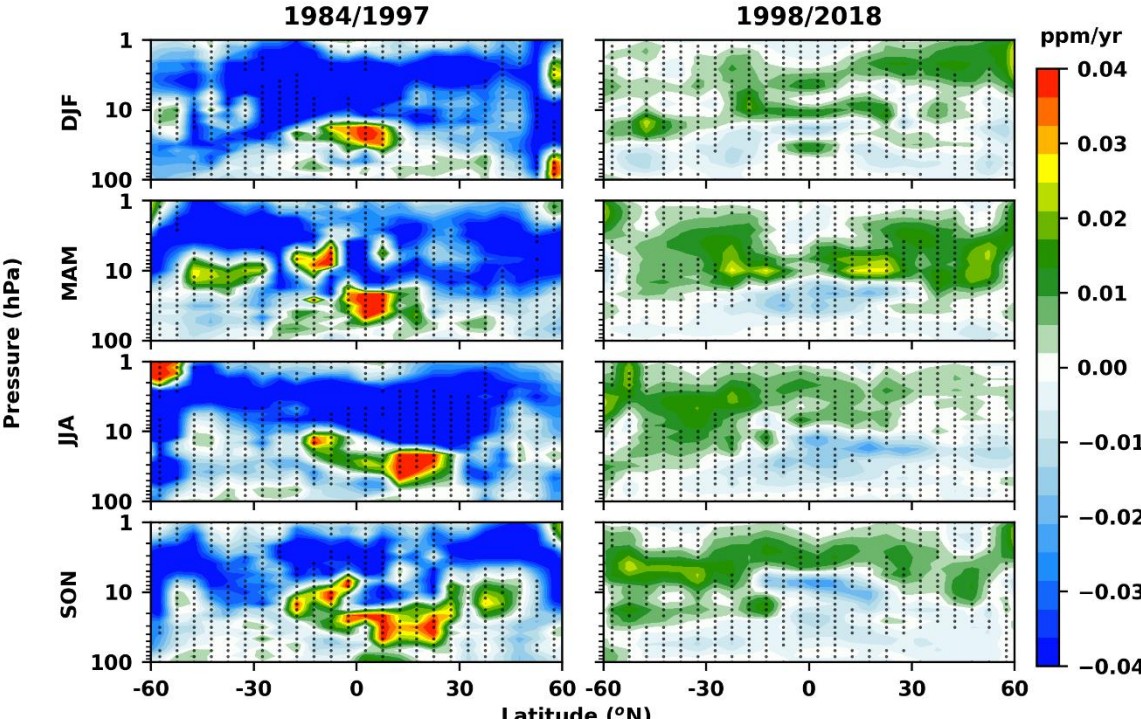

Figure 4: Trends in mixing ratio of ozone estimated for each season using the SWOOSH data for the periods of 1984–1997 and 1998–2018.
The hatched regions are statistically insignificant at 95% CI level. Here, DJF is December, January and February; MAM is March, April and
May; JJA is June, July and August; SON is September, October and November.

The GOZCARDS data also show similar trends, but the values are slightly higher (0.03–0.04 ppm/yr) in the middle and upper
stratosphere in both the northern and southern mid-latitudes. The main difference is the high statistically significant positive
trends in the northern mid- and high latitudes in JJA and southern mid and high-latitudes in DJF during 1984–1997 in
GOZCARDS. Similarly, there is a high significant negative trend in the southern high-latitude middle stratosphere in JJA and
northern high latitude middle stratosphere in DJF, and southern high latitude upper stratosphere in DJF and northern high-
latitude upper stratosphere in JJA, about 0.03–0.04 ppm/yr in GOZCARDS during the period 1998–2018. Therefore, we
have examined the difference between GOZCARDS and SWOOSH ozone, which is shown in Fig. S2. In general, GOZCARDS
shows relatively higher values in the middle stratosphere (25–35 km) until 2004, which is the Aura MLS period. However,
GOZCARDS shows slightly lower values with SWOOSH during the HALOE period, from 1991 to 2004, within 0.5 ppbv.
The agreement between both data sets is excellent in the lower and upper stratosphere, and throughout the stratosphere in
2004–2020, within 0.1 ppbv. The pre-1998 difference in the middle stratosphere is also reflected in the trend analysis, as
GOZCARDS show significant positive trends in DJF and MAM in the southern, and SON and JJA in the northern mid- and
high latitudes. Our results are consistent with those of Szelag et al. (2020), as they also find statistically significant negative



trends in the tropical lower stratosphere (up to -3%/yr), but positive trends in the middle and upper stratosphere in spring and summer.

We examined another dataset, the MERRA-2 reanalysis (Fig. S3), to find the trends derived from SWOOSH and GOZCARDS, and they replicate the information deducted from the previously discussed datasets. The trends are statistically nonsignificant in both periods, except in the topmost altitudes in the upper stratosphere. The trends computed with MERRA-2 data show slightly higher values (0.03–0.05 ppm/yr in the middle stratosphere and 0.02–0.03 ppm/yr in the lower stratosphere) at all altitudes in both periods. The negative trends in polar regions and positive trends in the tropical and southern middle latitudes are stronger than those found with SWOOSH, and are within ±0.03 ppm/yr. Also, the middle and lower latitude trends are slightly higher in the reanalysis data for the 1980–1997 period. The positive trends in the upper stratosphere and negative trends in the middle stratosphere around 10 hPa are stronger in the 1998–2018 period. Nonetheless, the trends in the tropical and mid-latitude lower stratosphere are positive (0.01–0.03 ppm/yr) or neutral in the MERRA-2 data, but are mostly neutral in SHOOWSH/GOZCARDS for the post-1998 period. However, note that, all these trends are nonsignificant. On the other hand, the trends estimated with TCO show either zero or negative up to -1.0 ± 0.6 DU/yr during the 1984–1997 period, except in the equatorial region in DJF and MAM, where it shows positive trends of 0.25 ± 0.50 DU/yr (Fig. S4). All trends show positive values in the next two decades (1998–2018), with their highest in the southern high latitudes of about 1.0 ± 0.2 DU/yr. However, the TCO trends are either zero or slightly negative in the tropical and mid-latitude regions (0.25–0.5 DU/yr). The highest negative trends are found in the northern high latitude of around 100° E in MAM and -180° W in SON.

The analyses with ERA-5 also yield similar trend estimates as for the MERRA-2 data (Fig. S5), but the values are slightly higher (0.1–0.2 ppm/yr, depending on altitude and latitude) in the ERA 5 data, particularly in the post-1997 period. The highest trend values in the southern and northern high latitudes are also significant in the ERA 5 data, but nonsignificant in other regions at most altitudes. The trends estimated with the TCO data show consistently higher values in ERA 5 as compared to MERRA-2 at most latitudes, about 0.5–1 DU/yr (Fig. S6). In all data sets, it is shown that the trends estimated in the tropical latitudes are either positive or neutral, and are nonsignificant.

### 3.2 Tropical ozone variability and trends

Our analyses (Fig. 4) show that there was substantial ozone loss in the 1984–1997 period at all latitudes and seasons, which is consistent in all data sets. The ozone trends are highly negative for Antarctica and smaller but also negative for the Arctic in the same period. Trends in midlatitude ozone are neutral or slightly negative. Neutral $O_3$ trends are also obtained for the tropics. This is consistent in both ozone profile and TCO measurements. On the other hand, the trends computed for the post-1997 period show positive values in Antarctica in most seasons, and in the upper stratosphere. Therefore, the analysis is in agreement with our current understanding of ozone chemistry in the stratosphere and with previous studies (Nair et al., 2015; Ball et al., 2019; Szelag et al., 2020; Sofieva et al., 2021; WMO, 2022).



Recently Lu (2022) claimed that there is strong ozone loss that he refers to as an "ozone hole" in the tropics in the past decades (1990–2000), which is reported to be present in all seasons and increasing in size day by day. The author further argues that this "ozone hole" is similar to that in Antarctica and even the chemical mechanisms causing it were the same. However, there are serious concerns about the study and the so-called tropical "ozone hole". First, the data Lu (2022) used are mainly from

the pre-satellite era and these data have plenty of gaps in the tropical region (Chipperfield et al., 2022). For instance, Figure 5 shows the data used by Lu (2022), in which there are large data gaps in the tropical latitudes in all three decades (1960, 1970 and 1980). These data gaps are in the middle stratosphere for 1960 and 1980, but in the entire lower and middle stratosphere for 1970. The ozone values in the tropical and middle latitudes are about 20–40 ppbv, and the largest values are in the northern polar regions, about 60–120 ppbv. From 1980 to 1990–2000, ozone increased (10–20 ppbv) across the latitudes, although it is

very small in the tropics. In contrast, ozone values decreased again in the tropical latitudes (mostly within 40 ppb), but increased in the middle and high latitudes at most altitudes. Note that there is no signature of an ozone hole in the Antarctica in this data, which also illustrates the problem of TOST data in accurately representing stratospheric ozone. In brief, very small values are observed in the TOST data in the tropics and the data gaps make it not suitable for statistical analysis. Second, the low ozone value region in the tropics is known to the scientific community for long (London, 1992) and the reason for this is the tropical

upwelling branch of BDC that carries low ozone air to the lower stratospheric altitudes (10–20 km).

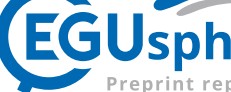

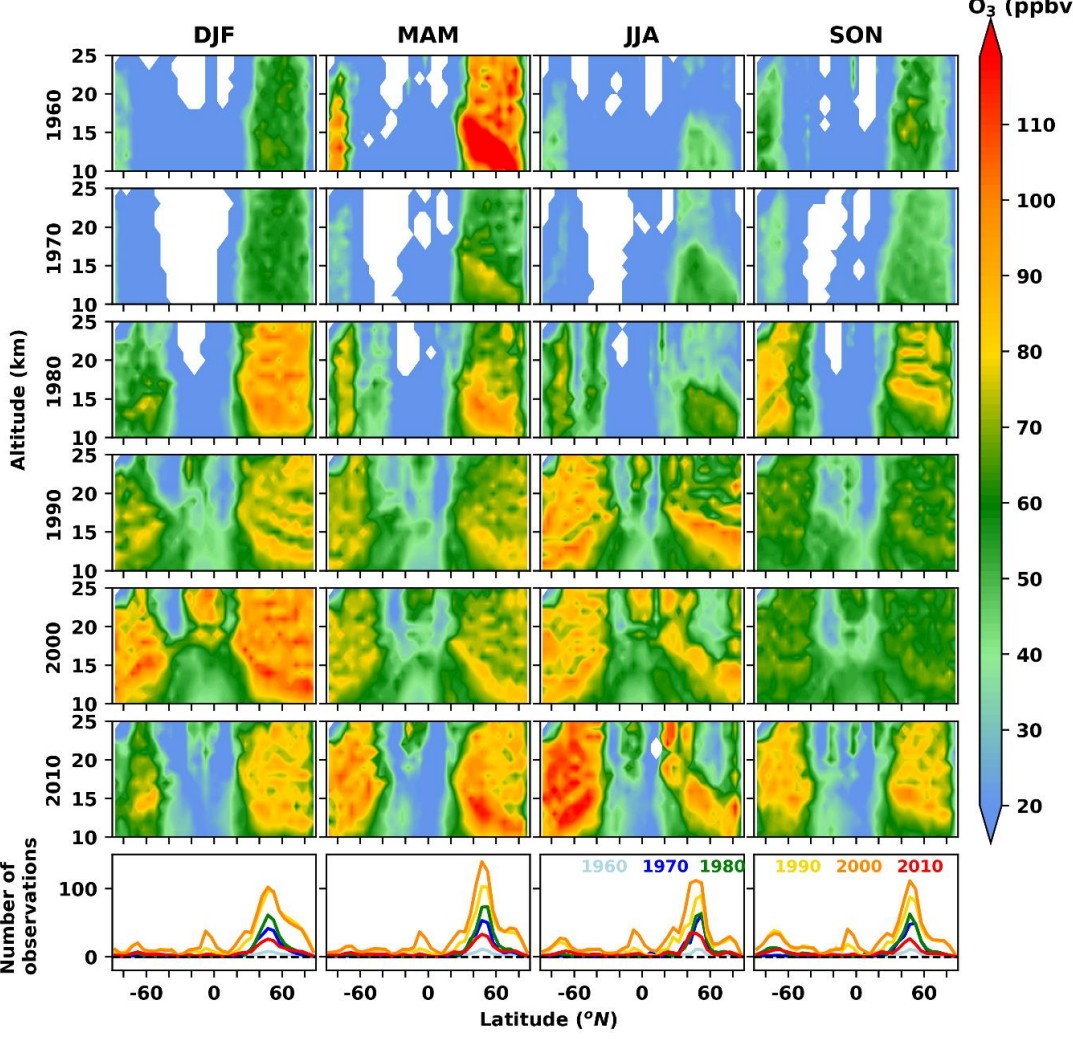

Figure 5: Average of vertical distribution of ozone from the Trajectory-mapped Ozonesonde dataset for Troposphere and Stratosphere (TOST) in each decade from 1960 to 2010. White areas indicate data gaps. Here, DJF is December, January and February; MAM is March, April and May; JJA is June, July and August; SON is September, October and November. The bottom panel shows the number of ozonesonde

observations at 19 km for each decade.

We have used all ozonesonde measurements available in the tropics from WOUDC to further examine the ozone values there (Fig. 6). As expected very small values are observed in the lower stratosphere, about 2 ppm. The decadal change of ozone is also very small (middle panel) in the past four decades, and the long-term analysis shows nonsignificant trends, at about 0.01 ± 0.008 ppm/yr for all three latitude bands (0–30° N, 0–30° S and 30°–30° N/S). In addition, the analysis with CAMS data for

the period of 2003–2020 shows very similar values in terms temporal evolution, decadal changes and long-term trends in the



tropical ozone. The trend values of CAMS ozone show nonsignificant positive trends of about $0.0018 \pm 0.002$ ppm/yr in the lower stratosphere (Fig. S7).

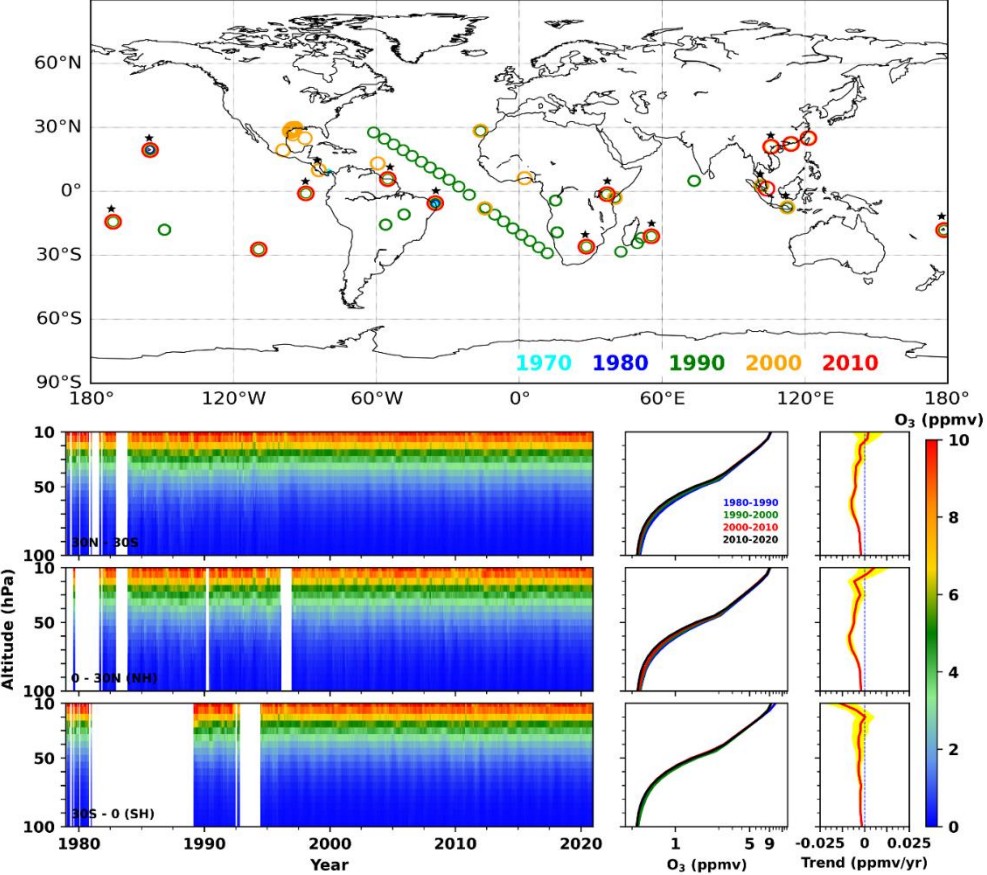

Figure 6. Top panel: Locations of the ozonesode stations in the tropics. The Southern Hemisphere ADditional OZonesondes (SHADOZ)
stations are marked with a black star (Bottom Left Panel): Ozone profiles from WOUDC ozonesonde averaged for the tropics, NH and SH, (Bottom Middle Panel) Mean distribution over each decade for each regions (Bottom Right Panel) and the yearly averaged trends for the period 1980–2020.

Furthermore, we have collocated the SHADOZ measurements to the nearest grids of the TOST data and estimated the linear trends and bias of TOST at 15–35 km. The decadal mean of SHADOZ data does not show any significant change in ozone
concentrations, except above 30–32 km, which can also be due to balloon measurement errors at these altitudes (Fig. S8). The trend estimated for the individual SHAODOZ stations exhibit either significant positive trends of about $0.01 \pm 0.005$ ppmv/yr or significant negative trends of about $0.01-0.035$ ppm/yr in the lower stratosphere (below 25 km). The middle stratospheric trends are positive at the southern hemispheric stations. The bias in TOST data, that were used by Lu (2022), estimated using



these collocated SHADOZ measurements in the tropics are shown in Fig. 7. Our analysis shows that the values of ozone in the

troposphere are reasonable, but there is a low bias of about 1–5 ppbv in TOST compared to the SHADOZ ozonesonde measurements, which is one of the reasons for the very low ozone values found in the study by Lu (2022). In addition, the comparison between TOST and satellite reanalysis data (GOZCARDS and SWOOSH) shows that TOST is biased low by 0.1–0.45 ppmv in the lower stratosphere, which increases with altitude (Fig. S9). Also, the ozone transported vertically from the tropical tropopause to the stratosphere usually is characterised by very low ozone values. Therefore, the low tropical ozone

values are driven by dynamics (Telford et al., 2009; Chipperfield et al., 2018).

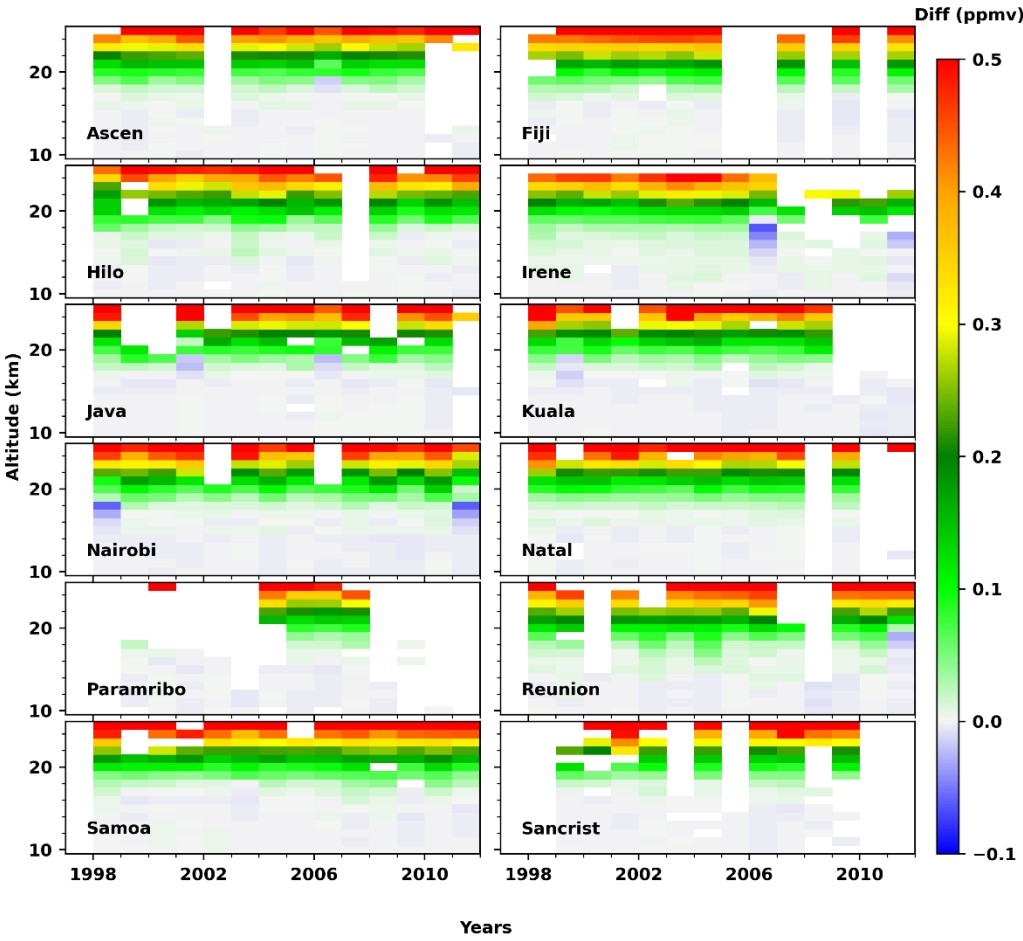

Figure 7: The bias in TOST dataset (SHADOZ - TOST in ppmv) calculated using collocated measurements from SHADOZ data for the years 1998–2012. SHADOZ is Southern Hemisphere ADditional OZonesondes and TOST is Trajectory mapped Ozonesonde dataset for the Stratosphere and Troposphere.



Third, Lu (2022) used the percentage change in ozone to define the "ozone hole", which is not a good metric to show how much ozone is present in a region. Rather, an ozone hole definition (i.e. ozone values below 220 DU) should be based on the amount of ozone present in a region, not relative to some other decade or a period. Apart from that, ozone loss is a seasonal process in the polar regions and therefore, the comparison must be made with respect to the period of ozone loss with respect to its starting year. In addition, the impact of "ozone hole" is depending on the amount of ozone present, not the amount relative

to previous decades in that region. Fourth, the amount of TCO in the tropical region was never below 220 DU and there is a slight increase in ozone in the stratosphere and troposphere after the year 2005 (see Fig. 3). No ozone measurements show values below 220 DU, but all depict a small increase in ozone after 2005, in contrast to the claim made by Lu (2022). Five, the formation of the polar vortex and of PSCs are key to ozone loss in the polar winter and spring. Formation of PSC particles are also required for the cosmic-ray-driven electron-induced reaction mechanism (CRE) mechanism put forward by Lu (2022).

However, no such phenomena are reported for the tropical stratosphere; indeed, there is no evidence for particles in the tropical stratosphere in measurements (Zou et al., 2022; Chipperfield et al., 2022). Therefore, no such heterogeneous ozone loss is observed in the low latitudes and there is no basis for the CRE theory (Grooß and Müller, 2011).

### 3.3 Reasons for the lower values of ozone in the Tropics

We also replicated the analysis made by Lu (2022), in addition to a detailed analysis by Chipperfield et al. (2022) with the
same TOST data, and find the following issues with Lu's claim on tropical ozone loss. (i) The TOST data Lu (2022) used are sparse in the tropical latitudes in the troposphere and stratosphere in all three previous decades of 1960s, 1970s and 1980s (see Fig. 5, top three panels and Fig. S10). Although the values are very small, which is expected (20–30 ppb) there, the data cannot be subtracted from another data set with gaps in them. One cannot claim any scientific process with interpolated data with huge gaps in them, as shown here. (ii) As opposed to Lu's statement of continuous decline, we find a slight increase in ozone
from 1980 to the next decades in various independent data sets.

The tropical stratospheric ozone has increased at least by 10–20 ppb in the past decades, in contrast to Lu's claim that, the so-called tropical "ozone hole" were expanding. However, the recent strengthening of BDC has reduced the ozone values in the tropical stratosphere, which is reflected in the analysis of ozone for recent decades (Butchart et al., 2006). Due to the accelerated motion of air in the tropics, the time for photochemical production of ozone is reduced, which is another reason for the declining
trend in ozone there (Avallone and Prather, 1996). The enhanced ozone transport to the middle latitudes further reduces ozone in the lower stratosphere (Wargan et al., 2018). In addition to the changes in relative strength of upper and lower branches of BDC (Butchart et al., 2006; Keeble et al., 2018; Abalos et al., 2019), the increase in halogen containing short-lived species as there are no regulations or polices to curb them (Hossaini et al., 2015), widening of extratropical troposphere (Zubov et al., 2013; Bognar et al., 2022), increased aerosol loading (Andersson et al., 2015), and unexpected emissions of CFC-11 (Fleming
et al., 2020) and inorganic iodine (Cuevas et al., 2018; Karagodin-Doyennel et al., 2021) could also decrease tropical lower stratospheric ozone. There is also a study suggesting that the reduction in solar activity might reduce ozone in the tropical



regions (Arsenovic et al., 2018). However, trend detection in the tropical latitudes is difficult due to the large dynamical variability there, as also mentioned by Stone et al. (2018). Note that the warming of tropical upper troposphere causes a sharp temperature gradient between tropics and mid-latitudes, which would push the jet, and thus lift the tropopause. This, in turn,
produces enhanced meridional transport between the regions (tropics to mid-latitudes) through the lower branch of BDC, and is projected to continue through the turn of the century. Henceforth, the tropical lower stratospheric ozone is also expected to decline further in the coming decades (Zubov et al., 2013). In brief, the change in tropical ozone presented in Lu (2022) is mostly due to the issues in the data used in his study, and the lower values of ozone in the troposphere are driven by dynamics. In the tropics, there are no new ozone loss processes and certainly, there is no "ozone hole" formed, as claimed.

Apart from these arguments, the claim by Lu (2022) regarding the lower ozone values and its impact is based on the volume (molar) mixing ratios in the tropical lower stratosphere. However, note that the ozone peak is around 30–35 km in these latitudes when we consider volume mixing ratios (molar mixing ratios), and thus, analyses of Lu (2022) miss the major part of tropical ozone. When we examine the column values, they are never below 220 DU and there is no big threat from UV radiation. Lu (2022)'s claim is solely with respect to a decadal dataset, which has only four profiles in each decade (see Fig.
S10) and the data set is available only for the lower stratosphere. On the other hand, here we have analysed a set of satellite, balloon-borne ground-based and reanalysed data to examine tropical ozone, and find that the claims are not properly based on measurements or model simulations, and the data Lu (2022) used are inadequate to analyse tropical stratospheric ozone. In addition, there is no such threat as Lu (2022) claimed due to the slight negative trends in ozone in the past two decades (1998–2018) as these changes are driven by stratospheric dynamics.

**4. Conclusions**

The analyses of stratospheric ozone in the tropics presented here show a consistent picture of ozone evolution in the past four decades. There is no significant loss or increase of tropical stratospheric ozone although slightly negative trends are found during the period of 2000–2020. Recent studies have suggested that the negative trends in the tropical upwelling region are caused by dynamical processes; including the increase in the speed of BDC. This is clearly pictured in the time series of
tropical ozone in recent years. The long-term trend in tropical TCO for the period (1998–2018) also shows no notable difference from the past decades. In summary, there is no tropical "ozone hole" and the evidence provided by Lu (2022) for such a phenomenon is seriously flawed.

*Data availability.* TOST data is available via https://woudc.org/archive/products/ozone/vertical-ozone-
profile/ozonesonde/1.0/tost/, GOZCARDS and MERRA−2 data are available on https://disc.gsfc.nasa.gov/, ERA 5 data are available on https://cds.climate.copernicus.eu/cdsapp#!/dataset/reanalysis-era5-single-levels?tab=overview, SHADOZ is



available via https://tropo.gsfc.nasa.gov/shadoz/, OMPS TCO is available at https://ozonewatch.gsfc.nasa.gov/, WOUDC data are available: https://woudc.org/home.php, SBUV MOD is available at https://acd-ext.gsfc.nasa.gov/Data_services/merged/, SWOOSH data is available at https://csl.noaa.gov/groups/csl8/swoosh/, TROPOMI data is available at https://sentinel.esa.int/web/sentinel/user-guides/sentinel-5p-tropomi, GSG data is available at http://www.iup.uni-bremen.de/UVSAT/datasets/merged-wfdoas-total-ozone

*Authorship contributions.* JK: Conceptualization, Methodology, Validation, Formal analysis, Investigation, Resources, Data Curation, Writing - Original Draft, Writing - Review & Editing, Visualization. GSG: Methodology, Software, Validation, Formal analysis, Investigation, Data Curation, Visualization, Writing – Original Draft. RM, SGB, JB: Formal analysis, Data Curation, Writing - Review & Editing.

*Competing interests.* JK, RM, and SGB are the editors of ACP, otherwise, there is no competing interest.

*Acknowledgements.* We thank the Chairman, CORAL and the Director, Indian Institute of Technology Kharagpur, for providing the facility for this study. The authors thank M. P. Chipperfield, Anne Thompson and L. Froidevaux for their comments and suggestions on the draft.

*Financial Support.* This study did not receive any project funding.

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
