# Peer review of "No severe ozone depletion in the tropical stratosphere in recent decades"

_EGUsphere, 2023_

## Referee Comment (RC1)

Review-10.5194-egusphere-2023-2574

This paper presents an analysis of trends in stratospheric ozone for several widely used ozone profile datasets as well as total ozone column datasets and reanalysis data with a primary focus on the tropical stratosphere. The paper is a response to Lu [2022] who claimed to have found a significant tropical stratospheric "ozone hole" in one particular and not widely use stratospheric ozone profile dataset. Such an extraordinary claim – inconsistent with current understanding - requires a thorough analysis beyond just one arbitrary chosen dataset. Lu [2022] does not provide such in depth analysis of other data, hence why this paper fills an important void.

The paper convincingly shows that the findings in Lu [2022] are not supported by any other data and datasets. The main reason is the incompleteness of the TOST dataset used in Lu [2022] (TOST) which is inadequate for long term tropical stratospheric ozone trend analysis prior to the 1990s.

While diving deeper into the matter, use of the TOST data for the analysis presented in Lu [2022] should have been a no-no for anyone looking into the papers describing and using the TOST data. Figure 12 of Liu et al. [2013; 10.5194/acp-13-11441-2013] copied below should have been a stark and obvious reminder that the tropics lack sufficient data in the 1970s and 1980s in the TOST data. Hence why Liu et al. [2013] in their analysis using TOST data do not analyze tropical trends prior to the 1990s.

Liu et al. [2013] contains more statements signaling that great care must be taken in the tropics and for the period prior to the 1990s:

> Abstract: "The agreement is better in the Northern Hemisphere, where there are more ozonesonde stations, than in the Southern Hemisphere; it is also better in the middle and high latitudes than in the tropics where reanalysis winds are less accurate."

> "Noting that there are large gaps in the tropics as ozone values were taken only from the grid cells where data are available for all four decades …"

> "The mean is area-weighted and is computed by taking values only from grid cells where ozone data are available for all four decades. Therefore, the global means miss large areas in the tropics (see Fig. 12)."

> "The distribution of ozone at 19.5 km is shown by decade from the 1970s to the 2000s in Fig. 12. The large gaps in the tropics and in the middle latitudes of the Southern Hemisphere in the 1970s are largely filled in by the SHADOZ program." (referee note: SHADOZ fills these gaps from the 1990s onwards)

> Conclusions: "Overall, we have more confidence in this climatology over the Northern Hemisphere than over the Southern Hemisphere, and in the middle and high latitudes than in the tropics."

Furthermore, the efforts to construct the TOST dataset dates back prior to 2010 using a computationally efficient and straightforward method to fill gaps in ozone data. That was a valuable advancement at the time given in particular the lack of methods for more advanced data assimilation techniques. However, the TOST approach has since been superseded by significant advances in both more advanced data assimilation techniques and computational power as evidenced by the MERRA-2 and ERA-5 reanalysis data. Hence TOST data soon will be outdated completely and its use likely will become obsolete. Anyone

using TOST data should at least acknowledge that and present additional support for findings based on TOST data.

Returning back to the paper under considering, overall it is well and logically organized and the findings are well supported by the data and analyses and suitable for publication in ACP. However, it appears the paper appears to be written in a bit of a hurry so there is some work to be done as outlined in the detailed comments below.

[Figure]

**Fig. 12.** Decadal variation in ozone at 19.5 km for **(a)** the 1970s, **(b)** the 1980s, **(c)** the 1990s, and **(d)** the 2000s. White areas indicate missing data.

Figure 12 from Liu, J., Tarasick, D. W., Fioletov, V. E., McLinden, C., Zhao, T., Gong, S., Sioris, C., Jin, J. J., Liu, G., and Moeini, O.: A global ozone climatology from ozone soundings via trajectory mapping: a stratospheric perspective, Atmos. Chem. Phys., 13, 11441–11464, https://doi.org/10.5194/acp-13-11441-2013, 2013.

**General:**

- "i.e." and "e.g." should be in italics.

- Southern Hemisphere should be in capitals.

- check consistent use of either ERA-5, ERA 5 or ERA5 (now all three are used)

**Specific comments, suggestions, typos**

Abstract, line 16: change to "the amount of column ozone is relatively small"

Abstract, line 17: add the range of TOC values at "high and mid-latitudes" (NH 275-425 DU; SH 275-350 DU; see for example Coldewey-Egbers et al. [2020] 10.5194/amt-13-1633-2020).

Abstract, line 18-19. Change to "No observational evidence was found of indications or signatures of severe stratospheric ozone depletion ..."

Abstract, line 20-21. Change to "Finally, current understanding and observational evidence does not provide any support for the possibility of an ozone hole occurring outside Antarctica today with …"

Introduction, line 26, add reference to Coldewey-Egbers et al. [2020]. Given what the paper is about it makes sense to add a recent reference in conjunction with London (1992).

Introduction, line 26-27. Change to "The production of ozone is effective at low latitudes hence ozone mixing ratios at middle and high latitudes are smaller …"

Introduction, line 28-30. The main reason why ozone columns increase is that atmospheric transport (Brewer-Dobson circulation) at high(er) latitudes is vertically downward. That causes air with similar mixing ratios to be to vertical levels with higher pressures and thus higher densities, even if mixing ratios remain the same or decrease, while the stratospheric column increases in geometrical thickness way from the tropics (tropical tropopause is at 16-18 km, outside of the tropics it is on average around 10 km). Somehow this should be reflected here.

Introduction, line 33. Change to "has shown a dramatic seasonal decrease"

Introduction, line 34, suggest to change to "Understanding of stratospheric ozone chemistry, model simulations" … "ozone loss theories" is a bit vague.

Introduction, line 37. "e.g." should be in italics.

Introduction, line 38. Change to "... deepened in the 1980s and peaked ..."

Introduction, line 39. Change to "... Antarctic lower stratosphere ..."

Introduction, line 43-52. This needs a makeover:

In contrast, Arctic stratospheric temperatures are relatively high and the polar vortex is frequently disturbed by planetary waves formed by the interaction of upper tropospheric winds, orography and land-sea contrasts, manifested as sudden stratospheric warmings (SSWs). The lack of persistent cold temperatures restricts Arctic stratospheric chlorine activation and ozone loss *(e.g.* Solomon, 1999) and causes strong year to year variability (von der Gathen et al., 2021) with generally column ozone staying

above 220 DU. Over Antarctic, on the other hand, springtime column ozone frequently falls below 220 DU. A weaker Brewer-Dobson circulation leads to lower stratospheric ozone amounts. In addition, a stronger, more persistent and much colder stratospheric vortex generally favors rapid springtime catalytic ozone destruction. Hence why the 220 DU column ozone threshold is widely used for characterizing the Ozone Hole. Such low values have been present …" … … Ozone loss in other regions, including the Arctic, never reach similar and widespread low levels of ozone like during Antarctic spring. Note that occasionally localized atmospheric dynamics can result in short lived small areas with low column ozone, so-called mini ozone holes (McCormack and Hood, 1997; James, 1998; Millán, L. F. and Manney, 2017).

*McCormack, J. P., & Hood, L. L. (1997). The frequency and size of ozone "mini-hole" events at northern midlatitudes in February. Geophysical research letters, 24(21), 2647-2650.*

*James, Pi M. "A climatology of ozone mini-holes over the Northern Hemisphere." International Journal of Climatology: A Journal of the Royal Meteorological Society 18.12 (1998): 1287-1303.*

*Millán, L. F. and Manney, G. L.: An assessment of ozone mini-hole representation in reanalyses over the Northern Hemisphere, Atmos. Chem. Phys., 17, 9277–9289, https://doi.org/10.5194/acp-17-9277-2017, 2017.*

Introduction, line 53: change to "Long term analyses show that column ozone loss …"

Introduction, line 62, reference to Weber et al. (2005) in relation to stable TCO values since 2000 should be replaced with reference to Weber et al. (2018 and/or 2022, both in ACP). Weber et al. (2005) is hardly relevant for post year-2000 conditions.

Introduction, lines 66-67. I do not understand the reference to Godin-Beekman et al. (2022) here. Does it mean that that paper like Bognar et al. (2022) shows a 1-3% reduction in lower stratospheric ozone since 2000? Or do you something else? Please clarify and modify the section accordingly.

Introduction, line 77. Change to "Chipperfield et al. (2022) in response showed that …"

Introduction, line 82, change to "Southern Hemisphere" (capitals)

Introduction, line 83. What "reprocessing" is referred to here?  SHADOZ? And if so, does that mean that before the reprocessing SHADOZ data was not good enough? Please clarify.

Introduction, line 85. Possibly replace "thorough" with "in-depth".

Section 2, data: given the importance of exploring a range of ozone datasets a table with estimated errors/precisions (in %) would be useful (for limb also as a function of altitude). If not, errors/precisions are missing GOZCARDS, WOUDC-ECC, TOST, MERRA-2 and ERA5 and should be added.

Section 2.1, line 96. Change to "ozone mixing ratios and standard errors … hPa and in 10° latitude bins."

Section 2.1, line 97-98. Change to "... ozone measurements. More details can ..."

Section 2.1, line 99. Change to "... are based on measurements of limb-sounding ..."

Section 2.1, lines 102-103. Change to "... by applying corrections calculated ..."

Section 3.1, line 187. Change to "... from the satellite dataset GOZCARDS."

Section 3.1, line 189. I don't know what is meant here with "effective". Probably what is meant is that the tropical stratosphere is a region of net ozone production where as middle to high latitudes are regions of net ozone destruction. Please clarify.

Section 3.1, line 191. minimum => minimal

Section 3.1, line 193. Change to the 3-month nomenclature (DJF, MAM, JJA, SON) rather than seasons in the tropics middle-to-high latitude seasonality is not useful to describe tropical seasonality. Furthermore, also later there are frequent references to particular seasons but given the inverse seasonality between NH and SH preferably use the 3-month nomenclature (DJF, MAM, JJA, SON) which, depending on what is described, could be augmented with its season. Otherwise it is confusing to read as mentally one continuously has to think "which months are what season in which hemisphere"

Section 3.1, line 206. "very low values in the southern hemisphere spring and autumn". I assume what is meant is spring and summer? SH ozone is not "very low" during autumn. Furthermore, probably better to write "very low values in the SH spring and low values in SH summer".

Section 3.1, line 223. "... no substantial loss in the tropics." => add the time period for which this statement is made and valid.

Section 3.1, line 225-227. Trend values for reanalysis data and the GSG data is missing. Possibly a table summarizing the various trends in various time periods/seasons/altitudes would be beneficial.

Section 3.1, line 237. Change to "We have also estimated ..."

Section 3.1, line 239. What is meant with "high statistically nonsignSeificant"? Please clarify.

Section 3.1, lines 260 & 262. Both lines mention differences in ppbv, but shouldn't that be ppmv?

Section 3.1, line 276. Remove comma in "note that, all these trends ..."

Section 3.1, line 277: replace "show" with "are"

Section 3.1, line 288. Change to "MAM, where trends are positive (0.25 +/- ...; Fig. S4)."

Section 3.1, line 288. "all datasets" ... please clarify which datatsets are meant here. Presumably all datasets discussed in the this section?

Section 3.1, lines 283 to 285. It appears there is an inconsistency with the previously used "ERA-5" as here it is "ERA 5" under the assumption this is not a print problem at my side.

Section 3.1, lines 286-287. Change to "In all datasets estimated post-1997 trends at tropical latitudes are either ..."

Section 3.2, line 296. "past decades (1990-2000)" should be "(1990-2020)"

Section 3.2, line 309. Add the following:

Third, Lu [2022] incorrectly assigns tropical altitudes above 10 km to the stratosphere whereas the tropical troposphere extends to 16-18 km [Seidel et al., 2001] and where very low ozone concentrations

can be found over the tropical Pacific due to vertical transport of clean tropical Pacific boundary layer air by convection [Kley et al., 1996]. Lu [2022] thereby incorrectly claims that Polvani et al. [2017] and Newton et al. [2018] report very low ozone values in the tropical lower stratosphere. Polvani et al. [2017] only discusses ozone at 70 hPa (18 km) and higher while Newton et al. [2018] assigns the low ozone observations to "uplift of almost-unmixed boundary-layer air" to altitudes of 100-150 hPa (14-17 km).

*Kley, D., Crutzen, P. J., Smit, H. G. J., Vömel, H., Oltmans, S. J., Grassl, H., & Ramanathan, V. (1996). Observations of near-zero ozone concentrations over the convective Pacific: Effects on air chemistry. Science, 274(5285), 230-233.*

*Seidel, D. J., Ross, R. J., Angell, J. K., and Reid, G. C. (2001), Climatological characteristics of the tropical tropopause as revealed by radiosondes, J. Geophys. Res., 106(D8), 7857–7878, doi:10.1029/2000JD900837.*

Section 3.2. end of section, add: "And finally, already more than two decades ago it was well established that - based on all available observational data - trends in tropical stratospheric ozone were largely absent or minimal at best for the period 1979-1997 [Staehelin et al., 2001], something neither acknowledged nor discussed in Lu [2022]."

*Staehelin, J., Harris, N. R. P., Appenzeller, C., and Eberhard, J. (2001), Ozone trends: A review, Rev. Geophys., 39(2), 231–290, doi:10.1029/1999RG000059.*

Section 3.2, line 316. Delete "there"

Section 3.2, line 318. about => approximately

Section 3.2, lines 320-321. In the tropical ozone => in tropical (lower) stratospheric ozone

Section 3.2, line 333. Change to "at Southern Hemispheric stations."

Section 3.2, line 348. Change to "in polar regions and therefore the comparison"

Section 3.2, line 352. Change to "No TCO measurements show"

Section 3.2, line 353. Change to "Formation of PSC particles is"

Section 3.3, line 362. Change to "are very small (20-30 ppb), which is expected there, the data"

Section 3.3, line 366. Change to "in the past decades according to our analysis of a wide range of available data."

Section 3.3, line 367. Delete the "However", just start the sentence with "The recent strengthening"

Section 3.3, line 378. mentioned => noted

Section 3.3, line 381. Change to "Henceforth, tropical lower stratospheric"

Section 3.3, line 384. Remove the comma in "and certainly there is no"

Section 3.3, line 386. Change to "However, the peak in ozone is around 30-35 km at these"

Section 3.3, line 387. Change to "Hence, the analyses of Lu (2022) miss"

Section 3.3, line 389. Change to "is solely based on one decadal dataset which has only four profiles"

Figure 2 caption. Change to "TCO less than 220 DU"

Figure 4. Since most of the trends are statistically insignificant, I think it would be better for the hatching to reflect the statistically significant trends. Hatching generally is used to identify statistically significant trends. Alternatively a second panel could be included showing only the significant trends. Same for Figure S1, S3, S4, S5 and S6.

Figure S8. Hatching of statistically significant trends is difficult if not impossible to discern. Maybe provide a second plot with only the statistically significant trends. Also, the figure caption should read "... ozonesonde profiles for different periods … at the 95% CI level."

---

## Author Comment (AC1)

**RESPONSE TO REFEREE #1 COMMENTS**

Thank you very much for your time, review and comments on MS. Please find answers to specific comments below. Please find the response to queries and the corresponding change in the MS in **blue typeface**. We do hope that the referee will find the revised version more interesting and recommend a publication very soon.

**General:**
- "i.e." and "e.g." should be in italics.
- Southern Hemisphere should be in capitals.
- check consistent use of either ERA-5, ERA 5 or ERA5 (now all three are used)

  Done. We have used these, as suggested, throughout the MS (e.g. SH, ERA-5, *e.g.*).

**Specific comments, suggestions, typos**
- Abstract, line 16: change to "the amount of column ozone is relatively small"

  Done. Please find it in line 15.

- Abstract, line 17: add the range of TOC values at "high and mid-latitudes" (NH 275-425 DU; SH 275-350 DU; see for example Coldewey-Egbers et al. [2020] 10.5194/amt-13-1633-2020).

  Done. Please find it in line 16.

- Abstract, line 18-19. Change to "No observational evidence was found of indications or signatures of severe stratospheric ozone depletion …"

  Done. Please find it in line 17.

- Abstract, line 20-21. Change to "Finally, current understanding and observational evidence does not provide any support for the possibility of an ozone hole occurring outside Antarctica today with …"

  Done. Please find it in line 19.

- Introduction, line 26, add reference to Coldewey-Egbers et al. [2020]. Given what the paper is about it makes sense to add a recent reference in conjunction with London (1992).

  Done. Please find it in line 25.

- Introduction, line 26-27. Change to "The production of ozone is effective at low latitudes hence ozone mixing ratios at middle and high latitudes are smaller …"

  Done. Please find it in lines 25–26.

- Introduction, line 28-30. The main reason why ozone columns increase is that atmospheric transport (Brewer-Dobson circulation) at high(er) latitudes is vertically downward. That causes air with similar mixing ratios to be to vertical levels with higher pressures and thus higher densities, even if mixing ratios remain the same or decrease, while the stratospheric column increases in geometrical thickness way from the tropics (tropical tropopause is at 16-18 km, outside of the tropics it is on average around 10 km). Somehow this should be reflected here.

  Done. Some parts here are removed to shorten it, and are now not in the revised MS.

- Introduction, line 33. Change to "has shown a dramatic seasonal decrease"

Done. Please find it in line 32.

- Introduction, line 34, suggest to change to "Understanding of stratospheric ozone chemistry, model simulations" ... "ozone loss theories" is a bit vague.
  Done. Please find it in line 33.

- Introduction, line 37. "e.g." should be in italics.
  Done. Please find it in line 34.

- Introduction, line 38. Change to "... deepened in the 1980s and peaked ..."
  Done. Please find it in line 37.

- Introduction, line 39. Change to "... Antarctic lower stratosphere ..."
  Some sections of the MS are omitted to reduce its length, as suggested by the referee. Therefore, this section is no longer part of the revised MS.

- Introduction, line 43-52. This needs a makeover:
  In contrast, Arctic stratospheric temperatures are relatively high and the polar vortex is frequently disturbed by planetary waves formed by the interaction of upper tropospheric winds, orography and land-sea contrasts, manifested as sudden stratospheric warmings (SSWs). The lack of persistent cold temperatures restricts Arctic stratospheric chlorine activation and ozone loss (e.g. Solomon, 1999) and causes strong year to year variability (von der Gathen et al., 2021) with generally column ozone staying above 220 DU. Over Antarctic, on the other hand, springtime column ozone frequently falls below 220 DU. A weaker Brewer-Dobson circulation leads to lower stratospheric ozone amounts. In addition, a stronger, more persistent and much colder stratospheric vortex generally favors rapid springtime catalytic ozone destruction. Hence why the 220 DU column ozone threshold is widely used for characterizing the Ozone Hole. Such low values have been present ..." ... ... Ozone loss in other regions, including the Arctic, never reach similar and widespread low levels of ozone like during Antarctic spring. Note that occasionally localized atmospheric dynamics can result in short lived small areas with low column ozone, so-called mini ozone holes (McCormack and Hood, 1997; James, 1998; Millán, L. F. and Manney, 2017).
  McCormack, J. P., & Hood, L. L. (1997). The frequency and size of ozone "mini-hole" events at northern midlatitudes in February. Geophysical research letters, 24(21), 2647-2650.
  James, Pi M. "A climatology of ozone mini-holes over the Northern Hemisphere." International Journal of Climatology: A Journal of the Royal Meteorological Society 18.12 (1998): 1287-1303.
  Millán, L. F. and Manney, G. L.: An assessment of ozone mini-hole representation in reanalyses over the Northern Hemisphere, Atmos. Chem. Phys., 17, 9277–9289, https://doi.org/10.5194/acp-17-9277-2017,2017.
  Done. These are mentioned in lines 41–43.

- Introduction, line 53: change to "Long term analyses show that column ozone loss ..."
  Done. Please find it in line 46.

- Introduction, line 62, reference to Weber et al. (2005) in relation to stable TCO values since 2000 should be replaced with reference to Weber et al. (2018 and/or 2022, both in ACP). Weber et al. (2005) is hardly relevant for post year-2000 conditions.
  Done. Please find it in line 47.

- Introduction, lines 66-67. I do not understand the reference to Godin-Beekman et al. (2022) here. Does it mean that that paper like Bognar et al. (2022) shows a 1-3% reduction in lower stratospheric ozone since 2000? Or do you something else? Please clarify and modify the section accordingly.
  Done. Please note that we have now specified the values are regions with both references in lines 51-52.

- Introduction, line 77. Change to "Chipperfield et al. (2022) in response showed that ..."
  Done. Please find it in line 61.

- Introduction, line 82, change to "Southern Hemisphere" (capitals)
  Done. Please find it in line 66.

- Introduction, line 83. What "reprocessing" is referred to here? SHADOZ? And if so, does that mean that before the reprocessing SHADOZ data was not good enough? Please clarify.
  Done. Please find it in lines 67–69.

- Introduction, line 85. Possibly replace "thorough" with "in-depth".
  Done. Please find it in line 73.

- Section 2, data: given the importance of exploring a range of ozone datasets a table with estimated errors/precisions (in %) would be useful (for limb also as a function of altitude). If not, errors/precisions are missing GOZCARDS, WOUDC-ECC, TOST, MERRA-2 and ERA5 and should be added.
  Done. We have included the errors attached with each dataset in the methods section.

- Section 2.1, line 96. Change to "ozone mixing ratios and standard errors ... hPa and in 10° latitude bins."
  Done. Please find it in lines 82–83.

- Section 2.1, line 97-98. Change to "... ozone measurements. More details can ..."
  Done. Please find it in line 85.

- Section 2.1, line 99. Change to "... are based on measurements of limb-sounding ..."
  Done. Please find it in line 86.

- Section 2.1, lines 102-103. Change to "... by applying corrections calculated ..."
  Done. Please find it in line 89.

- Section 3.1, line 187. Change to "... from the satellite dataset GOZCARDS."
  Done. Please find it in line 172.

- Section 3.1, line 189. I don't know what is meant here with "effective". Probably what is meant is that the tropical stratosphere is a region of net ozone production where as middle to high latitudes are regions of net ozone destruction. Please clarify.
  Done. We have used "higher" instead of effective there. Please find in it line 174.

- Section 3.1, line 191. minimum => minimal
  Done. Please find it in line 177.

- Section 3.1, line 193. Change to the 3-month nomenclature (DJF, MAM, JJA, SON) rather than seasons in the tropics middle-to-high latitude seasonality is not useful to describe tropical seasonality. Furthermore, also later there are frequent references to particular seasons but given the inverse seasonality between NH and SH preferably use the 3-month nomenclature (DJF, MAM, JJA, SON) which, depending on what is described, could be augmented with its season. Otherwise it is confusing to read as mentally one continuously has to think "which months are what season in which hemisphere"

  Done. This is done throughout the MS.

- Section 3.1, line 206. "very low values in the southern hemisphere spring and autumn". I assume what is meant is spring and summer? SH ozone is not "very low" during autumn. Furthermore, probably better to write "very low values in the SH spring and low values in SH summer".

  Done. Please find it in line 191.

- Section 3.1, line 223. "... no substantial loss in the tropics." => add the time period for which this statement is made and valid.

  Done. Please find it in line 204.

- Section 3.1, line 225-227. Trend values for reanalysis data and the GSG data is missing. Possibly a table summarizing the various trends in various time periods/seasons/altitudes would be beneficial.

  Done. Please find it the table in Supplementary file, Table S1.

- Section 3.1, line 237. Change to "We have also estimated ..."

  Done. Please find it in line 219.

- Section 3.1, line 239. What is meant with "high statistically nonsingnficant"? Please clarify.

  Done. High trend values, but nonsignificant. This is rephrased in line 221.

- Section 3.1, lines 260 & 262. Both lines mention differences in ppbv, but shouldn't that be ppmv?

  Done. Yes, Please find it in lines 239 and 241.

- Section 3.1, line 276. Remove comma in "note that, all these trends ..."

  Done. Rephrased the sentence.

- Section 3.1, line 277: replace "show" with "are"

  Done. Rephrased the sentence.

- Section 3.1, line 288. Change to "MAM, where trends are positive (0.25 +/- ...; Fig. S4)."

  Done. Rephrased the sentence in lines 241-243.

- Section 3.1, line 288. "all datasets" ... please clarify which datatsets are meant here. Presumably all datasets discussed in this section?

  Done. Yes, please find it in lines 246-247.

- Section 3.1, lines 283 to 285. It appears there is an inconsistency with the previously used "ERA-5" as here it is "ERA 5" under the assumption this is not a print problem at my side.

  Done. We have made it uniform everywhere now, as ERA-5. For instance, lines 205, 213.

- Section 3.1, lines 286-287. Change to "In all datasets estimated post-1997 trends at tropical latitudes are either ..."
  Done. We have paraphrase the sentence, please find it in lines 238–239.

- Section 3.2, line 296. "past decades (1990-2000)" should be "(1990-2020)"
  Done. Please find it in line 253.

- Section 3.2, line 309. Add the following:
  Third, Lu [2022] incorrectly assigns tropical altitudes above 10 km to the stratosphere whereas the tropical troposphere extends to 16-18 km [Seidel et al., 2001] and where very low ozone concentrations can be found over the tropical Pacific due to vertical transport of clean tropical Pacific boundary layer air by convection [Kley et al., 1996]. Lu [2022] thereby incorrectly claims that Polvani et al. [2017] and Newton et al. [2018] report very low ozone values in the tropical lower stratosphere. Polvani et al. [2017] only discusses ozone at 70 hPa (18 km) and higher while Newton et al. [2018] assigns the low ozone observations to "uplift of almost-unmixed boundary-layer air" to altitudes of 100-150 hPa (14-17 km).
  Kley, D., Crutzen, P. J., Smit, H. G. J., Vömel, H., Oltmans, S. J., Grassl, H., & Ramanathan, V. (1996). Observations of near-zero ozone concentrations over the convective Pacific: Effects on air chemistry. Science, 274(5285), 230-233.
  Seidel, D. J., Ross, R. J., Angell, J. K., and Reid, G. C. (2001), Climatological characteristics of the tropical tropopause as revealed by radiosondes, J. Geophys. Res., 106(D8), 7857–7878, doi:10.1029/2000JD900837.
  Done. Please find it in lines 308–314.

- Section 3.2. end of section, add: "And finally, already more than two decades ago it was well established that - based on all available observational data - trends in tropical stratospheric ozone were largely absent or minimal at best for the period 1979-1997 [Staehelin et al., 2001], something neither acknowledged nor discussed in Lu [2022]."
  Staehelin, J., Harris, N. R. P., Appenzeller, C., and Eberhard, J. (2001), Ozone trends: A review, Rev. Geophys., 39(2), 231–290, doi:10.1029/1999RG000059.
  Done. Please find it in lines 320–323.

- Section 3.2, line 316. Delete "there"
  Done.

- Section 3.2, line 318. about => approximately
  Done. Please find it in line 268.

- Section 3.2, lines 320-321. In the tropical ozone => in tropical (lower) stratospheric ozone
  Done. Please find it in line 268.

- Section 3.2, line 333. Change to "at Southern Hemispheric stations."
  Done. Please find it in line 289.

- Section 3.2, line 348. Change to "in polar regions and therefore the comparison"
  Done. Please find it in line 305.

- Section 3.2, line 352. Change to "No TCO measurements show"

Done. Please find it in line 314.

- Section 3.2, line 353. Change to "Formation of PSC particles is"
  Done. Please find it in line 316.

- Section 3.3, line 362. Change to "are very small (20-30 ppb), which is expected there, the data"
  Done. Please find it in line 327.

- Section 3.3, line 366. Change to "in the past decades according to our analysis of a wide range of available data."
  Done. Please find it in lines 331–332.

- Section 3.3, line 367. Delete the "However", just start the sentence with "The recent strengthening"
  Done. Please find it in line 332.

- Section 3.3, line 381. Change to "Henceforth, tropical lower stratospheric"
  Done. Please find it in line 348.

- Section 3.3, line 384. Remove the comma in "and certainly there is no"
  Done. Please find it in line 350.

- Section 3.3, line 386. Change to "However, the peak in ozone is around 30-35 km at these"
  Done. Please find it in line 352.

- Section 3.3, line 387. Change to "Hence, the analyses of Lu (2022) miss"
  Done. Please find it in line 353.

- Section 3.3, line 389. Change to "is solely based on one decadal dataset which has only four profiles"
  Done. Please find it in line 355.

- Figure 2 caption. Change to "TCO less than 220 DU"
  Done. Please find it the revised Figure 1.

- Figure 4. Since most of the trends are statistically insignificant, I think it would be better for the hatching to reflect the statistically significant trends. Hatching generally is used to identify statistically significant trends. Alternatively a second panel could be included showing only the significant trends. Same for Figure S1, S3, S4, S5 and S6.
  Done, we have re-drawn the figures, for instance, Figures S1, S3, S4 etc., with statistically significant trends stippled. Please find the new figures in the supplementary file.

- Figure S8. Hatching of statistically significant trends is difficult if not impossible to discern. Maybe provide a second plot with only the statistically significant trends. Also, the figure caption should read "… ozonesonde profiles for different periods … at the 95% CI level."
  Done, we have redrawn the figure with statistically significant trends. Please find the new Figure S10 in the revised MS.

---

## Author Comment (AC2)

**RESPONSE TO REFEREE #2 COMMENTS**

This paper reports on stratospheric ozone trends in the tropical region and is motivated by the recent claim by Lu (2022a, 2002b) that a tropical ozone hole exists. A rebuttal to the paper by Lu was published by Chipperfield et al. (2022) that argued that this claim is not substantiated by observations and that the interpretation of galactic cosmic rays (GCR) being a cause for this severe loss is highly unlikely. This paper confirms that there are only small tropical ozone changes observed in the tropics over the last decades supporting the arguments given in the rebuttal by Chipperfield et al. So far so good. My issue with the paper is that it is not clear to me what new insights are provided here. The small trends in the tropics have been reported before (WMO 2022, Weber et al. 2022, Godin-Beekmann et al. 2022). I also do not see what new arguments are delivered here with respect to what has been already reported by Chipperfield et al. in response to Lu (2022). The authors need to state clearly what new aspects are brought in here that justifies publication of this paper. A major revision is required.

Thank you very much for your time, review and comments on MS. Please find answers to specific comments below. Please find the response to queries and the corresponding change in the MS in blue typeface. We do hope that the referee will find the revised version more interesting and recommend a publication very soon.

Please note that we have used more measurements, reanalyses and different statistics measures for delineating the trends and interpreting the results. In addition, we have also compared our results with previous studies, including Chipperfield et al., Please find the answers to specific comments below.

**Other major issues**
* Various ozone and reanalysis data are used in this study and trends calculated from them. Apparently not all datasets have been updated to end of 2022. MERRA-2, ERA5, GSG, and SBUV/OMPS MOD data are available up to end of 2022. Stratospheric ozone trends from GOZCARDS, SWOOSH, and reanalysis data are only shown up to 2018, why not up to end of 2022?

Done. We have extended the data up to 2022. However, GOZCARDS data are available only up to 2021.

* To make the paper more concise (and probably shorter), the authors should try to focus on trends in the tropical region (see paper title), rather than global and extratropical regions, which are extensively discussed here.

Done, we have shortened the paper, focusing on the global tropical region. We hope that the referee would find the revised MS more interesting.

* The regression model used is poorly described. It is pretty much standard that ozone trends are derived using multiple linear regression (MLR) that contain additional terms (proxies) describing ozone variability (e.g. WMO 2022, Weber et al. 2022, Godin-Beekmann et al. 2022). What is the justification for not using MLR? A comparison between MLR (previous work) and simple linear regression results (this work) may become difficult.

Done, now we use the GOZCARDS and SWOOSH datasets to regress the ozone at different levels of the stratosphere. Please find the new figure, Figure S7, in the revised Supplementary file. Furthermore, a detailed discussion of the MLR model and results are also added to the revised MS. Please find it in lines 271–278.

---

## Editor Decision (ED1)

**Editor comments on Manuscript No egusphere-2023-3574**

*No severe ozone depletion in the tropical stratosphere in recent decades by Kuttipurath et al.*

P1, L15: "in the tropics" should be moved before "during the past five decades".

P2, L29: downwards -> propagate downwards

P2, L31: earth -> Earth (?) . Later you indeed start with a capital letter. However, if "Earth" or "earth" is the correct version should be checked with the ACP guidelines and then corrected accordingly.

P2, L27: Mccromick -> McCormick

P3, L71: I would write here "polar stratospheric clouds" (even if they are in the tropics not polar, but then the reader at least knows what type of clouds are needed). Alternative you could also write what kind/type of PSCs are required.

P3, L71: Which CFC-12 observations. Please clearly state which CFC-12 observations you refer to (which instrument/model).

P3, L79ff: I would add a "the" before each long version of the instrument names (even if this is often skipped when only the abbreviation is used).

P3, L80: I would suggest to add "its successor" before SAGE II

P3, L81: add "respectively, " after Aura.

P4, L86: Add "the" before "Stratospheric Water and…..".

P4, L86: Mention that this is also a merged data set.

P4, L97: "(/2)" correct? Or is there a typo?

P4, L101: Merged -> merged

P4, L102ff: Also here, I would suggest to add "the" before the long names of the instruments.

P4, L113: location of SHADOZ -> location of the SHADOZ

P5, L117: radiosonde -> radiosondes

P5, L128: regions -> region

P5, L129: Hyplit -> HYSPLIT

P5, L130: of TOST -> of the TOST

P5, L134: Add "The" before "Tropospheric Ozone….."

P5, L134: in earth's -> in the earth's (Earths's?)

P5, L136: "and the detection of ozone depletion events" -> detects ozone depletion events (?)

P5, L137: Add "the" -> the GODFIT

P5, L139: Add "The" -> The Ozone Mapping……

P5, L139: Add "the" -> on-board the Suomi…..

P5, L140: delete "in"

P6, L144: Add "The" -> The Ozone Monitoring……

P6, L145: Add "the" -> scattered by the

P6, L149: Add "the" -> of the global

P6, L160: add "The" ->  The Ozone Mapping

P6, L163: hereby -> here and what do you mean with explanatory variables? Rephrase?

P7, L173: add "merged" -> the merged satellite data

P7, Figure 1: O3 in the legend cut.

P8, Figure 1 caption: add "the" twice and add "s" to show ->……. by the Ozone Mapping shows the region of the ozone hole……

P9, Figure 2 caption: Since you already have introduced the instrument names in the text, this is obsolete to be repeated in the figure captions.

P9, L221: data set or dataset? Please check the manuscript, I had the following that you used both versions of writing it.

P9, L229: MAM not introduced. I would suggest to write March, April and May (MAM).

P10, L234: better to write "the trends" instead of "that"?

P10, L235: the trends …is -> either the trends…..are or the trend …..is

P10, L236: Same here, mixture of singular and plural.

P10, L237: check sentence. Rather "and these in the lower and middle stratosphere are nonsignificant….

P10, L239: Aura MLS? Rather UARS MLS since Aura MLS was launched at around 2004 and not before.

P11, L255: I would suggest to write 1960s, 1970s and 1980s instead of 1960, 1970 and 1980.

P11, L262: low ozone air -> better to write "air with low ozone"?

P13, L277: ppm/yr -> ppm/yr-1 (see ACP guidelines)

P13, Figure 5 caption: In many cases capital letters are used where small letters should be used, e.g. Bottom Right/Left Panel -> bottom left/right panel

P14, L289: I think for ACP it should rather be written with small letters -> southern hemispheric.

P14, L292: Sonde -> sonde

P14, L293: bias -> biases

P15, Figure 6: Since the abbreviation of the instruments have been introduced in the text, this does not need to be repeated in the figure caption.

P16, L318: "mechanism" twice -> one obsolete

P16, L319: add "ice" -> for ice particles

P16, l322: is -> are

P16, L324: Tropics -> tropics

P17, L338: of BDC -> of the BDC

P17, L356: This is a bit of a repetition here and I would suggest to move this remainder of theis paragraph to the conclusion section.

P17, L358: reanalysed -> reanalyses

P17, L366: of BDC -> of the BDC

P18, L382: skip "the" and replace "is" by "are".

---

## Author Response (AR2)

**RESPONSE TO EDITOR CORRECTIONS**

Thank you very much for your time and comments on this MS. Please find answers to specific comments below. Please find the response to queries and the corresponding change in the MS in **blue typeface**. We do hope that the editor will find the revised version more interesting and recommend publication soon.

P1, L15: "in the tropics" should be moved before "during the past five decades".
Done, please find it in line 14.

P2, L29: downwards -> propagate downwards
Done, please find it in line 29.

P2, L31: earth -> Earth (?) . Later you indeed start with a capital letter. However, if "Earth" or "earth" is the correct version should be checked with the ACP guidelines and then corrected accordingly.
Done, changed it in line 31

P2, L27: Mccromick -> McCormick
Done, corrected it in line 38

P3, L71: I would write here "polar stratospheric clouds" (even if they are in the tropics not polar, but then the reader at least knows what type of clouds are needed). Alternative you could also write what kind/type of PSCs are required.
Done, rephrased it in line 72

P3, L71: Which CFC-12 observations. Please clearly state which CFC-12 observations you refer to (which instrument/model).
Done, its observations and model results. This is mentioned in line 73.

P3, L80: I would suggest to add "its successor" before SAGE II
Done, please find it in line 82

P3, L81: add "respectively, " after Aura.
Done, there is no sequence here,  so "respectively" is not added.

P4, L86: Mention that this is also a merged data set.
Done, please find it in line 89

P4, L97: "(/2)" correct? Or is there a typo?
Done, corrected it in line 100

P3, L79ff: I would add a "the" before each long version of the instrument names (even if this is often skipped when only the abbreviation is used). , P4, L86: Add "the" before "Stratospheric Water and.....".
P4, L102ff: Also here, I would suggest to add "the" before the long names of the instruments.
P5, L134: Add "The" before "Tropospheric Ozone.....", P5, L139: Add "The" -> The Ozone Mapping......

P6, L144: Add "The" -> The Ozone Monitoring......, P6, L160: add "The" -> The Ozone Mapping
Done, we have added in some places as appropriate, but not everywhere. Thank you.

P4, L101: Merged -> merged
Done, please find it in line 104

P4, L113: location of SHADOZ -> location of the SHADOZ
Done, please find it in line 116

P5, L117: radiosonde -> radiosondes
Done, please find it in line 120

P5, L128: regions -> region
Done, please find it in line 131

P5, L129: Hyplit -> HYSPLIT
Done, please find it in line 132

P5, L130: of TOST -> of the TOST
Done, please find it in line 133

P5, L134: in earth's -> in the earth's (Earths's?)
Done, , Please find it in line 137

P5, L136: "and the detection of ozone depletion events" -> detects ozone depletion events (?)
Done, please find it in line 140

P5, L137: Add "the" -> the GODFIT
Done, please find it in line 140

P5, L139: Add "the" -> on-board the Suomi.....
Done, please find it in line 142

P5, L140: delete "in"
Done.

P6, L145: Add "the" -> scattered by the
Done, please find it in line 148

P6, L149: Add "the" -> of the global
Done, please find it in line 152

P6, L163: hereby -> here and what do you mean with explanatory variables? Rephrase?
Done, please find it in line 166

P7, L173: add "merged" -> the merged satellite data
Done, please find it in line 176

P7, Figure 1: O3 in the legend cut.
Done, please find it in the revised Figure 1

P8, Figure 1 caption: add "the" twice and add "s" to show ->……. by the Ozone Mapping shows the region of the ozone hole……
Done, please find it in line 189

P9, Figure 2 caption: Since you already have introduced the instrument names in the text, this is obsolete to be repeated in the figure captions.
Done, please find it in lines 214–217

P9, L221: data set or dataset? Please check the manuscript, I had the following that you used both versions of writing it.
Done, we have consistently used "dataset", e.g. line 103, 126, 204, 218

P9, L229: MAM not introduced. I would suggest to write March, April and May (MAM).
Done, please find it in line 227

P10, L234: better to write "the trends" instead of "that"?
Done, it is already mentioned, which is why we have used "that of"

P10, L235: the trends …is -> either the trends…..are or the trend …..is
Done, please find it in line 232

P10, L236: Same here, mixture of singular and plural.
Done, corrected find it in line 234

P10, L237: check sentence. Rather "and these in the lower and middle stratosphere are nonsignificant….
Done, corrected it in line 235

P10, L239: Aura MLS? Rather UARS MLS since Aura MLS was launched at around 2004 and not before.
Done, please find it in line 237

P11, L255: I would suggest to write 1960s, 1970s and 1980s instead of 1960, 1970 and 1980.
Done, please find it in line 253

P11, L262: low ozone air -> better to write "air with low ozone"?
Done, please find it in line 259

P13, L277: ppm/yr -> ppm/yr-1 (see ACP guidelines)
Done, please find it in line 273

P13, Figure 5 caption: In many cases capital letters are used where small letters should be used, e.g. Bottom Right/Left Panel -> bottom left/right panel
Done, the sentences are complete. So capital letters are used. Please find it in line 280

P14, L289: I think for ACP it should rather be written with small letters -> southern hemispheric.
Done, it is abbreviated now in line 287

P14, L293: bias -> biases
Done, please find it in line 291

P15, Figure 6: Since the abbreviation of the instruments have been introduced in the text, this does not need to be repeated in the figure caption.
Done, please find it in line 297

P16, L318: "mechanism" twice -> one obsolete
Done, we have removed one, please find it in line 314.

P16, L319: add "ice" -> for ice particles
Done, please find it in line 315

P16, l322: is -> are
Done, please find it in line 319

P16, L324: Tropics -> tropics
Done, please find it in line 321

P17, L338: of BDC -> of the BDC
Done, please find it in line 335

P17, L356: This is a bit of a repetition here and I would suggest to move this remainder of this paragraph to the conclusion section.
Done, please find it in line 360

P17, L358: reanalysed -> reanalyses
Done, please find it in line 362

P17, L366: of BDC -> of the BDC
Done, please find it in line 358

P18, L382: skip "the" and replace "is" by "are"
Done, "is" for the "no competing interest". Thank you.